# Protective Effects of Choline against Inflammatory Cytokines and Characterization of Transport in Motor Neuron-like Cell Lines (NSC-34)

**DOI:** 10.3390/pharmaceutics14112374

**Published:** 2022-11-04

**Authors:** Sana Latif, Young-Sook Kang

**Affiliations:** Research Institute of Pharmaceutical Sciences, College of Pharmacy, Sookmyung Women’s University, 100 Cheongpa-ro 47-gil, Yongsan-gu, Seoul 04310, Korea

**Keywords:** amyotrophic lateral sclerosis, choline, NSC-34 cell lines, choline-like transporters, inflammatory cytokines, nerve growth factor

## Abstract

Choline, a component of the neurotransmitter acetylcholine, is essential for nervous system functions, brain development, and gene expression. In our study, we investigated the protective effect and transport characteristics of choline in amyotrophic lateral sclerosis (ALS) model cell lines. We used the wild-type (WT) motor neuron-like hybrid cell line (NSC-34/hSOD1^WT^) as a control and the mutant-type (MT; NSC-34/hSOD1^G93A^) as a disease model. The uptake of [^3^H]choline was time-, pH-, and concentration-dependent. [^3^H]Choline transport was sodium-dependent, and, upon pretreatment with valinomycin, induced membrane depolarization. Gene knockdown of Slc44a1 revealed that choline-like transporter 1 (CTL1) mediates the transport of choline. In NSC-34 cell lines, the specific choline transporter inhibitor, hemicholinium-3 demonstrated significant inhibition. Donepezil and nifedipine caused dose-dependent inhibition of [^3^H]choline uptake by the MT cell line with minimal half inhibitory concentration (IC_50_) values of 0.14 mM and 3.06 mM, respectively. Four-day pretreatment with nerve growth factor (NGF) resulted in an inhibitory effect on [^3^H]choline uptake. Choline exerted protective and compensatory effects against cytokines mediators. Hence, the choline transport system CLT1 may act as a potential target for the delivery of novel pharmacological drugs, and the combination of drugs with choline can help treat symptoms related to ALS.

## 1. Introduction

Neurodegeneration is a major problem that is caused by various genetic factors and environmental aspects such as nutrients [1]. Amyotrophic lateral sclerosis (ALS), which is a deadly disease, can cause motor neuron death leading to an inability to move due to muscle weakness and paralysis [2]. The exact pathophysiological mechanism of ALS is unknown; some related conditions include oxidative stress and mutated superoxide dismutase (SOD1) proteins. The SOD1 protein tends to aggregate and misfold, causing familial ALS [3]. Previous research has shown that the axon-enriched fraction of SOD1 was misfolded and accumulated at the onset of the disease [4]. In the spinal cord of transgenic mice, the kinesin-associated protein 3 (KAP3) subunit is connected to SOD1 and primarily in charge of transporting choline acetyltransferase (ChAT). Reduced ChAT transport activity due to the possession of KAP3 by misfolded SOD1 leads to the pathophysiology of ALS [5]. Recently, researchers have been in search of nutrients with beneficial and protective effects for ALS [6]. Researchers have found that vitamins and nutrients play disease-modifying roles in ALS treatment and can reduce the degeneration rate of motor neurons, reducing the disease progression rate [7]. Choline is obtained from various food and vegetables and is naturally produced by the liver. Free choline in the blood plasma membrane crosses the blood-brain barrier and is taken up by cholinergic nerve terminals in a high-affinity uptake of choline that is energy and sodium-dependent. This uptake system by which choline provided for the synthesis of ACh is transported into the neuron is the rate-limiting step for the synthesis of neurotransmitters [8]. Choline deficiency can lead to ailments such as atherosclerosis, liver disease, and various neurodegenerative diseases [9]. Another study has stated that choline is a methyl donor and micronutrient that is essential for the development and growth of a healthy brain. Choline helps in regulating brain cholinergic signals through acetylcholine synthesis [10]. Prior research has proposed that choline not only acts as a micronutrient but also as a neuroprotective agent through an epigenetic mechanism of action for the development of brain programs in the early stage of life [11]. 

Transport systems involved in choline transport can be divided mainly into three types. The first type is the organic cation transporters (OCTs), such as solute carrier transporters with the gene name (SLC22A1-2), which have low choline affinity and are sodium-independent [12]. The second type is commonly known as a choline transporter (CHT/SLC5A7) and is sodium-dependent, has high choline affinity, and is hemicholinium-3-sensitive. CHT is highly expressed in cholinergic neurons present at the synaptic cleft and terminals of neurons [13]. The third type, the choline-like transporter (CTLs/SLC44A1-5) has an intermediate affinity for choline and is sodium-independent [14]. In the blood-brain barrier (BBB), choline is transferred from the blood to the brain using a carrier-mediated mechanism whose characteristics were comparable to OCTs [15]. In addition, in a conditionally immortalized syncytiotrophoblast cell line (TR-TBT), choline was transported by CTL1 [16]. In this study, we performed a gene knockdown study for the identification of the transporters responsible for choline transport in NSC-34 cell lines. Additionally, we studied the effect of nerve growth factor (NGF) on the uptake of choline in ALS model cell lines. NGF protein plays a critical role in the development of sensory, sympathetic, and peripheral neurons, in addition to raising the ChAT activity [17]. 

In our earlier research, we studied how the transport of certain drugs and amino acids was altered in ALS model cell lines (NSC-34). In ALS patients, it has been demonstrated that the concentration of choline was lower compared with that in healthy controls, which is related to the impairment of upper motor neurons in ALS disease [18]. The exact cause of the low concentration of choline in ALS is not known. Therefore, in our study, we investigate the affinity and capacity of choline transport, as well as characteristics of the transporter in ALS model cell lines that will help in the transport of choline in the motor neuron that will, in turn, help in the synthesis of ACh. In addition, we also study the ability of choline to protect against inflammatory cytokines. Furthermore, cell viability via MTT assay and the effect of NGF pretreatment has been assessed in ALS model cell lines

## 2. Materials and Methods 

### 2.1. Radioisotope and Chemical Compounds

Radioactive isotope [^3^H]choline (78.3 Ci/mmol) was purchased from PerkinElmer (Boston, MA, USA). Choline chloride, acetylcholine (ACh), hemicholinium-3 (HC-3), verapamil, diphenhydramine (DPH), nifedipine, donepezil, edaravone, tetraethyl ammonium (TEA), amiloride, betaine, and L-carnitine were purchased from Sigma Aldrich (St. Louis, MO, USA). All chemicals were of reagent grade.

### 2.2. NSC-34 Cell Lines

Motor neuron-like hybrid cell lines (NSC-34) were provided by Prof. Hoon Ryu (KIST, Seoul, Korea). The NSC-34 cell lines were categorized as, either, control (wild-type [WT]; NSC-34/hSOD1^WT^) or disease model (mutant-type [MT]; NSC-34/hSOD1^G93A^) cell lines. The NSC-34 cell line is a fusion cell line derived from enriched spinal cord motor neurons and neuroblastoma [19]. Mouse NSC-34 cells are stably expressing human superoxide dismutase 1 (hSOD1, WT), which is considered as the control, and the mutant NSC-34/hSOD1G93A (MT) cell line, which overexpresses the human mutant SOD1 gene mutation with the substitution of glycine with alanine at position 93. This MT cell line displayed less differentiation and proliferation while mimicking the clinical circumstances linked to motor neuronal dysfunction in ALS. Therefore, NSC-34/hSOD1G93A MT is considered as the disease model cell line [20]. For the experimental purpose, WT cells of passage #4–19 and MT cells of passage #6–17 were used. 

### 2.3. Culture of NSC-34 Cell Lines

Collagen type-I-coated culture dishes were used for the culture and sub-culture of cells, and the cells were grown in Dulbecco’s modified Eagle’s medium (Invitrogen, San Diego, CA, USA) supplemented with 10% fetal bovine serum (Hyclone, Logan, UT, USA), 100 U/mL penicillin, and 100 µg/mL streptomycin (Invitrogen). The protocol from an earlier study was followed [21]. Rat tail collagen type-I-coated 24-well culture plates were prepared by the initial seeding of cells at a density of 1 × 10^5^ cells/well. The cells were incubated at 37 °C in a humidified atmosphere with 5% CO_2_ until reaching confluence [22]_._

### 2.4. Uptake Study of [^3^H]Choline on NSC-34 Cell Lines

The uptake study was conducted in accordance with previous research [23]. The extracellular fluid (ECF) buffer was prepared under the usual conditions: the pH was adjusted to 7.4, and the temperature was 37 °C. The cells were washed three times with 1 mL of ECF, and uptake was performed by adding 200 µL of ECF containing the radioisotope compound [^3^H]choline (31.9 nM) to each well for the selected time of 5 min at 37 °C, in either the presence or absence of a substrate or inhibitor. The uptake rate was calculated according to the formula described in a previous study [24]. 

### 2.5. Uptake Study under Sodium- and Potassium-Free Conditions

To assess the effect of sodium ions on the uptake of [^3^H]choline, ECF buffer was prepared by replacing sodium chloride with *N*-methyl-d-glucamine (NMG^+^) and lithium chloride (LiCl). The effect on the membrane potential was observed by the replacement of sodium ions with equimolar concentrations of potassium chloride (KCl) and KHCO_3_ in ECF, and uptake was performed after pretreatment with 10 µM valinomycin [22]. 

### 2.6. Kinetic Parameter Estimation of [^3^H]Choline Uptake in NSC-34 Cell Lines

In order to evaluate the concentration dependency, unlabeled choline samples of 0 to 1 mM were used, and the uptake of [^3^H]choline at 5 min was assessed. Further, the kinetic parameters, the Michaelis–Menten constant (K_m_), and the maximum uptake rate (V_max_) were calculated using Equation (1):V = [V_max1_·C/(K_m1_ + C) + V_max2_·C/(K_m2_ + C)] + K_d_·C(1)
where C is the concentration of the unlabeled choline, and V is the initial uptake rate of [^3^H]choline at 5 min [25]. Similarly, the half inhibitory concentration (IC_50_) was measured by calculating the percentage of inhibition in the uptake of choline, either in the presence or absence of amiloride, diphenhydramine, donepezil, and nifedipine. 

All graphs and bar charts were created, and statistical analyses were carried out with Sigma Plot (version 12, Systat Software Inc., Richmond, CA, USA). Each data point is expressed as the mean ± standard error mean (S.E.M). In addition, one-way analysis of variance (ANOVA) with Dunnett’s post-hoc test was used to evaluate the data. A two-tailed unpaired t-test was used to compare the two groups. Statistics were considered significant if the *p*-value was less than 0.05.

### 2.7. Real-Time PCR Analysis 

According to the manufacturer’s protocol, total RNA was extracted from the NSC-34 cell lines using the RNeasy mini kit (Qiagen, Valencia, CA, USA). A high-capacity RNA-to-cDNA kit was used to prepare cDNA from the total RNA using a gene amplification system (Mycycler; Bio-Rad Laboratories, Hercules, CA, USA). A Real-time PCR system (RT-PCR; Applied Biosystems, Foster City, CA, USA) was used for the analysis according to the instructions described by the manufacturer. CHT/Slc5a7 and CTL1/Slc44a1 expressions were normalized to that of GAPDH, a housekeeping gene. The StepOnePlus system (Applied Biosystems) was used for the analysis of the 48-well plates according to the protocol [23].

### 2.8. Cytokines & Nerve Growth Factor Study Methods 

ALS model cell lines were treated with tumor necrosis factor (TNF)-α and lipopolysaccharide (LPS) at a concentration of 20 ng/mL, either in the presence or absence of choline at a concentration of 1 mM, for 24 h. Then, an uptake analysis of [^3^H]choline was carried out. For the cell viability assay using MTT, the concentration for H_2_O_2_ pretreatment was selected as 300 µM. In the NGF study, the concentration of NGF was selected as 50 ng/mL, and cells were pretreated 2 and 4 days before the uptake analysis.

## 3. Results

### 3.1. Time Course and Effect of pH on [^3^H]Choline Transport in NSC-34 Cell Lines

To investigate the characteristics of choline transport in ALS model cell lines, an uptake study was performed using WT and MT cell lines. A time course study was performed and showed that, until 60 min at pH 7.4, uptake was time-dependent and linear until 15 min; however, the uptake rate was significantly lower at 5 min in the disease model cell line MT compared with that of the WT (Figure 1A). Therefore, for further study, we selected 5 min for further study. Next, to scrutinize the effect of extracellular pH on the uptake rate of choline, the pH of ECF was varied from acidic (6.4) to alkaline (8.4), and the analysis was conducted in comparison with the outcomes at physiological pH 7.4. The transport of choline was pH-sensitive in ALS model cell lines (Figure 1B). In both cell lines, the transport rate was significantly decreased at pH 6.4, whereas it was markedly increased at pH 8.4, in comparison with that at pH 7.4. At the same time, the pH-dependent transport rate of choline in the MT cell line was lower than that in the WT cell line (Figure 1B). 

### 3.2. Na^+^ and K^+^ ion Dependence on [^3^H]Choline Uptake in NSC-34 Cell Lines

The uptake of choline was performed in the presence of 10 mM of cold choline; it showed more than 90% of strong inhibition in both cell lines. Furthermore, choline uptake was influenced by the absence of sodium from the ECF buffer in both cell lines. NSC-34 cell lines were depolarized by removing Na^+^ in the buffer solution, and [^3^H]choline uptake was markedly decreased (Figure 2). In addition, the disruption of membrane potential by KCl resulted in a significant decrease in the uptake of choline in both cell lines. Hence choline transport in NSC-34 cell lines was voltage-sensitive (Figure 2).

### 3.3. Transport Kinetics Parameters of [^3^H]Choline in NSC-34 Cell Line

We analyzed the mechanism underlying the transport of choline in ASL model cell lines. The uptake of [^3^H]choline was performed with varying concentrations of cold choline, and increased uptake was observed with increasing concentrations. Hence, the transport of choline in NSC-34 cell lines was concentration-dependent (Figure 3).

The kinetic parameters indicated that, at the high-affinity site, the affinity in the MT cell line is significantly high compared to the WT line, whereas the capacity is significantly low in the MT cell line. At the low-affinity site, there was significantly high affinity in the MT cell line compared to the WT line; however, there was no significant difference in capacity. The Eadie–Hofstee plot (Figure 2, insets) produced two lines, suggesting the involvement of two saturation processes in the uptake of choline in NSC-34 cell lines.

### 3.4. RT-PCR Analysis of CTL1 Expression in ALS Model Cell Lines 

To identify which transporter was involved in the transport of choline in ALS model cell lines, gene knockdown was performed using Slc44a1 siRNA, and, thereafter, [^3^H]choline uptake was carried out. The results indicated that both cell lines transfected with Slc44a1 siRNA showed a marked decrease in the uptake of [^3^H]choline in comparison with those treated with the control non-targeted pool siRNA (Figure 4A). These results were further confirmed by analyzing the mRNA expression levels after transfecting the cell lines with Slc44a1-targeted and non-targeted siRNA. PCR analysis showed a significant decrease in the expression of CTL-1 after Slc44a1 siRNA transfection (Figure 4B). The uptake in the MT cell line was significantly lower compared to that in the WT cell line, which may be due to a reduced number of transporters in the MT cell line as can be inferred from Figure 4. Taken together, these findings indicate that CTL1 is involved in the uptake of choline in ALS model cell lines.

### 3.5. Inhibitory Effects of Pharmacological Compounds on Choline Transport by ALS Model Cell Lines

Inhibition studies were performed to determine the substrate selectivity of choline transport in NSC-34 cell lines (Figure 5). The data revealed that acetylcholine (ACh), a neurotransmitter, and HC-3, which is a potent choline uptake inhibitor, showed a marked decrease in the uptake of [^3^H]choline in the WT and MT cell lines. Cationic drugs such as DPH and paeonol caused a significant inhibition of [^3^H]choline uptake in both cell lines. Verapamil and nifedipine Ca^+^ channel blockers also significantly inhibited choline uptake in NSC-34 cell lines. In addition, quinidine (an antiarrhythmic drug), amiloride (a diuretic drug), and propranolol (a beta blocker) also significantly inhibited the uptake of choline in NSC-34 cell lines. A known substrate of the OCT transporter, TEA, markedly inhibited choline uptake, whereas betaine (a choline metabolite) and L-carnitine (the OCTN substrate) did not show any inhibitory effect in ALS mode cell lines (Figure 5). Further, edaravone (a neuroprotective drug) did not have any significant effect on the uptake of choline in ALS model cell lines.

### 3.6. IC_50_ of Drugs in Disease Model Cell Lines

Next, we investigated the effects and interactions of pharmacological drugs for their potential inhibitory effect on the uptake of choline in the MT cell line. Donepezil, a drug to treat dementia in neurodegenerative diseases, was selected. The concentrations used for the disease model cell line were 0.05–2 mM. IC_50_ values for amiloride and DPH were calculated as 1.04 µM and 61.0 µM, respectively. Moreover, donepezil and nifedipine showed a concentration-dependent inhibition on choline uptake in MT cells with IC_50_ values of 0.14 mM and 3.06 mM, respectively (Figure 6). Based on these results, it is inferred that the ALS disease model is sensitive to therapeutic drugs including amiloride, DPH, and donepezil.

### 3.7. Uptake of [^3^H]Choline Increases with NGF

NGF is a neurotrophin that is necessary for growth regulation and the survival of neurons. Pre-incubation of cells was performed in our investigation to observe the impact of NGF on ALS model cell lines. NGF pretreatment (50 ng/mL) was performed for 2 and 4 days prior to the uptake study. The results indicated that pre-exposure of cells to NGF for 2 days caused no significant difference in the uptake of choline. However, after 4 days of pretreatment, there was a marked decrease in [^3^H]choline uptake in both cell lines. (Figure 7). mRNA expression levels of CTL1 were analyzed after treating MT cells with NGF for 4 days. The finding suggested that the transporters were depleted, resulting in a decrease in the uptake of choline. Overall, these results showed that NSC-34 cell lines were sensitive to prolonged NGF treatment.

### 3.8. Effect of Choline on the Cytotoxicity Induced by LPS and TNF-α in ALS Model Cell Lines

Pre-incubation of the MT cell line with pro-inflammatory cytokines LPS and TNF-α for 24 h resulted in a significant increase in the uptake of [^3^H]choline, whereas the addition of 1 mM of choline to these pretreated MT cell line resulted in a restorative effect. MT cell line treated with cold choline only showed a marked increase in the uptake of [^3^H]choline as compared with that in non-treated control cells. A similar pattern was observed for the mRNA expression levels of CTL1, where the expression of CTL1 was increased with the inflammatory states, and the addition of choline resulted in restoration, demonstrating that choline reduces the cytotoxicity induced by the pro-inflammatory cytokines (Figure 8). Therefore, we hypothesized that the increase in the uptake of choline occurred due to the increase in the level of CTL1 transporter under the cytotoxic effect. 

### 3.9. Cell Viability Analysis by MTT Assay

To assess the cytotoxicity induced by cytokines, we performed an MTT assay. Initially, the MT cell line was exposed to LPS, TNF-α, and H_2_O_2_ for 24 h. Cell viability was visibly decreased in the disease model cell line, whereas the cells co-incubated with choline showed a marked increase in viability. These cells were subjected to the MTT assay, revealing similar results (Figure 9). The choline-only-treated cells showed no distinct variation from the non-treated control cells. 

## 4. Discussion

Choline is an important neurotransmitter and an essential micronutrient that has been found to have neuroprotective benefits [1]. Choline has been shown to delay the signs of aging and memory loss, which are frequently experienced by patients suffering from neurodegenerative diseases such as Alzheimer’s disease (AD) [10,11]. However, the characteristics of choline transport in ALS have not been studied; therefore, we aimed to elucidate the choline uptake characteristics in ALS model cell lines. Moreover, choline has shown neuroprotective effects in the brain and neurodegenerative diseases; therefore, in the current study, the protective role of choline was evaluated after pre-incubation of NSC-34 cell lines with inflammatory cytokines (LPS, TNF-α, and H_2_O_2_). 

In particular, an in vitro uptake study was used to explore the characteristics of choline transport, and we found that the uptake of [^3^H]choline was time-, pH-, and sodium-dependent in motor neuron-like NSC-34 cell lines (Figure 1 and Figure 2). An earlier study in brain capillaries has shown time dependency up to 60 s [26]; however, in both WT and MT NSC-34 cell lines, it was linear until 15 min (Figure 1A). Our results are concurrent with the study of choline transport in TR-TBT cells demonstrating the time- and pH-sensitivity; however, in TR-TBTs, the uptake was only slightly dependent on sodium, whereas, in NSC-34 cell lines, it was markedly reduced in the absence of sodium [16]. Additionally, [^3^H]choline uptake was voltage-sensitive in NSC-34 cell lines, as indicated by our studies on the disruption of membrane potential (Figure 2). A similar pattern was found in TR-TBT cells [16]. Moreover, our saturation kinetics study revealed that choline in motor neuron cell lines is transported in a concentration-dependent manner and, in contrast to the TR-BBB cell line [15], it has shown two distinct affinity sites in the ALS model cell lines (Figure 3). Kinetic parameters are similar to the transport of choline in rat brain microvessels reporting K_m_ at the high-affinity site to be 6.1 µM and V_max_ 10.6 pmol/mg protein/min [27], similar to our results suggesting that the affinity was 10.1 µM and capacity was 101 pmol/mg protein/min in the control (WT) ALS cell line at the high-affinity site (Table 1). Previous research on choline transport in disease models has also reported the comparison between the stroke-prone spontaneously hypertensive rats (SHRSP) and normotensive Wistar rats (WKY). The kinetics of choline showed that the capacity was significantly lower in the SHRSP rats as compared to WYK rats [28], similar to our results showing a lower capacity transport system in the MT cell line of ALS as compared to the WT control.

Next, we explored the transporter responsible for choline transport in motor neuron cell lines. From previous investigations in rats, mice, and humans, homologous genes of CTL1 have been discovered [13,29,30]. The CTL1 clone was also discovered in the species *Torpedo marmorata* [13]. CHT-1 was believed to be exclusive throughout the brain in cholinergic neurons [31]; however, there was no expression of CHT-1 in motor neuron cell lines in this study (data were not shown). CTL-1 was highly expressed in ALS model cell lines, therefore, the potential role of CTL-1 in the transport of choline was established by knocking down the Slc44a1 gene, and the results showed a significant decrease in the uptake of choline after siRNA transfection. The reduction of choline transport was due to reduced Slc441 transporter, as shown in Figure 4. Additionally, choline in TR-TBT was transported to a small extent by the sodium-dependent CTL1 transport system [16], and in microglial SIM-A9 cells, there was expression of both CTL-1,2 and transport of choline was sodium-independent [32]. In contrast to these findings, in our current study, choline uptake by CTL1 was strongly sodium-dependent. 

We also investigated how different pharmacological compounds affected the uptake of choline in NSC-34 cell lines. ACh is an important neurotransmitter synthesized within the cholinergic neurons, which sends the information through the nerve impulse from motor neurons to skeletal muscles [33]. In our study, ACh strongly inhibited the uptake of [^3^H]choline at a concentration of 2 mM in both cell lines. In addition, HC-3, which is a selective inhibitor of CHT, showed up to 50% inhibition in both cell lines. TEA, which is a major substrate of the OCT transporter [34], showed a slight inhibition in both cell lines. Verapamil is a calcium channel blocker and, in SOD1^G93A^ mice, has been shown to rescue the denervation of the skeletal muscle and aid the survival of motor neurons [35]. In our study, verapamil showed significant inhibition of choline uptake in both cell lines. Nifedipine, another Ca^+^ blocker, also showed significant inhibition of [^3^H]choline uptake in both cell lines. Our results are consistent with the uptake of choline in TR-TBT [16]. DPH, an antihistamine and cationic drug, has been shown to reduce the symptoms associated with ALS [36,37]. DPH caused a significant inhibition of the uptake of [^3^H]choline in NSC-34 cell lines. Amiloride is a potassium-sparing diuretic and, in our study, caused significant inhibition in both cell lines (Figure 5). Previous research has shown that amiloride along with riluzole reduces the volume of the brain, prevents the structural damage of axons, and acts as a neuroprotective agent [38]. Next, we assessed the inhibitory concentrations of several therapeutic drugs to increase their clinical usefulness in the disease state. Donepezil, amiloride, DPH, and nifedipine were selected for the IC_50_ study, and concentrations were calculated according to the inhibition of [^3^H]choline uptake by half in the disease model cell line (Figure 6). An earlier study has revealed through mechanistic analysis that donepezil can interfere with the actions of numerous proteins that are affected by SOD1 mutation in ALS [39].

A previous study has shown that differentiation by various combinations of cAMP, retinoic acid, and nerve growth factors (NGF) might provide cells of different morphologic maturity as well as activities of acetylcholine and acetyl-CoA metabolism [40] NGF is a neurotrophin with a particular cholinotropic role in the brain. It was discovered to increase the density of cholinergic M2 autoreceptors in the hippocampus, septum, and cortical areas having cholinergic neurons and their terminals in the axon [41]. NGF effects are mediated by specific high-affinity choline uptake expression in cultured neurons and tyrosine kinase-type receptors. Various studies have revealed the beneficial effects of NGF and its use in glaucoma, corneal ulcers, human cutaneous ulcers, and retinal maculopathy [42]. In the neuronal axons, phosphatidylcholine enzymes are present and are required for axonal growth [43]. Earlier research has reported that, in pheochromocytoma PC12 cells, NGF induced neurite outgrowth, resulting in phosphatidylcholine PtdCho synthesis [44]. In our study, we selected the concentration of 50 ng/mL based on a previous study reporting the effect of various concentrations of NGF with the maximum effect at 50 ng/mL in PC12 cells [44]. Our findings revealed that in both NSC-34 cell lines, the uptake of choline significantly decreased upon exposure to NGF for 4 days (Figure 7). Similar to our results, NGF has been shown to increase the level of choline incorporated into phosphatidyl-choline in PC12 cells at 4 days at the concentration of 50 ng/mL [44]. To confirm our results, we analyzed mRNA expression levels and obtained similar results, showing a decrease in the transporter (CTL1) activity in the MT cell line (Figure 7C). A previous study has shown that, under particular circumstances, cholinergic neurons treated with NGF show greater sensitivity to neurotoxic assaults [40]. Low-affinity p75 receptors are thought to be responsible for this phenomenon, and their activation has been frequently linked to an increase in cell death and modification of stress responses [45]. 

Additionally, we examined the protective effect of choline against inflammatory states. Several studies have stated that the immune and inflammatory responses are related to the pathology of ALS [46]. A human study reported that inflammatory cytokines were raised in the plasma of patients with ALS [47]. Therefore, we decided to determine the effects of LPS and TNF-α, with or without the addition of choline, on these inflammatory states in the disease model cell line. Upon pretreatment for 24 h, the inflammatory cytokines alone resulted in increased uptake and a subsequent increase in the mRNA expression of the CTL1 transporter in the MT cell line (Figure 8). However, the addition of choline at the inflammatory states resulted in reduced [^3^H]choline uptake and Slc44a1 mRNA expression. The results indicated that choline exerted a protective effect against these cytokines. A similar pattern of uptake was shown in previous studies of taurine and carnitine in ALS model cell lines, demonstrating the neuroprotective effects of these compounds against the inflammatory states [48,49]. Similar to our results the uptake of choline in M2 microglia has been shown to be increased by LPS [32]. Another research study on a mouse model for pain reported that, by stimulating the nicotinic receptor a7, choline decreases TNF release from the macrophages, and choline appears to be effective in the inhibition of inflammatory cytokines [50]. Furthermore, cell viability was examined after incubating the cells with the cytokine mediators, including LPS, TNF-α, and H_2_O_2_, inducing oxidative stress. The results revealed a notable decline in MT cell viability. The addition of choline under these inflammatory stress-induced conditions resulted in a compensatory effect (Figure 9). The results obtained are similar to those of the studies conducted on valproic acid and paeonol, showing the restorative effects of these compounds on NSC-34 cell lines [51,52]. Previous research has reported the antioxidant effect of choline in immune organs by regulating the antioxidant system and decreasing oxidative damage [53].

## 5. Conclusions

Overall, the results obtained in this study imply that the CTL1 transporter may help the sodium-dependent transport of choline into the neurons and will help in understanding the transport phenomenon. Long-term exposure to NGF has shown a sensitive effect in ALS model cell lines. Additionally, choline exerts a protective role against oxidative stress and excitotoxins. Taken altogether, our results suggest that choline may be used in combination with other drugs for alleviating the symptoms related to ALS. 

## Figures and Tables

**Figure 1 pharmaceutics-14-02374-f001:**
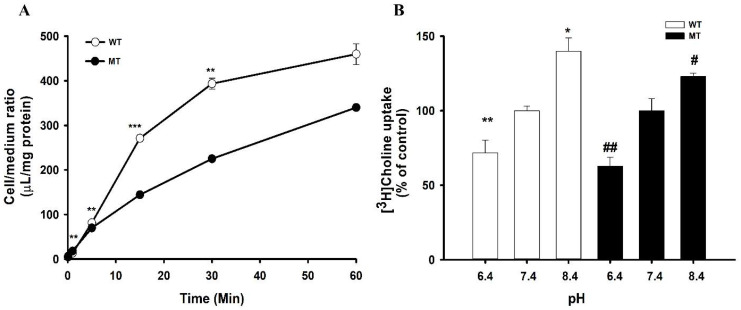
Time and pH dependency of [^3^H]choline uptake in ALS model cell lines. (**A**) The time course of [^3^H]choline uptake was observed at pH 7.4 at 37 °C for 0–60 min using ECF buffer solution in both cell lines. ** *p* < 0.01 and *** *p* < 0.001, significantly different from the WT (**B**) Uptake was observed using ECF solutions with different pH (6.4 to 8.4) for 5 min using 7.4 as a control in WT and MT cell lines. * *p* < 0.05 and ** *p* < 0.01, significantly different from pH 7.4 in WT; # *p* < 0.05, ## *p* < 0.01; significantly different from pH 7.4 MT. Each value represents the mean ± S.E.M (*n* = 3–4).

**Figure 2 pharmaceutics-14-02374-f002:**
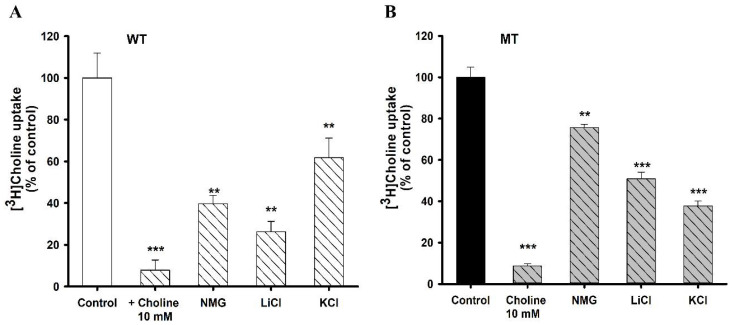
Ion dependency and membrane potential in (**A**) WT and (**B**) MT cell lines. [^3^H]Choline uptake was performed at pH 7.4 and 37 °C for 5 min in the presence or absence of sodium and potassium. Ten micromoles of valinomycin were used for pretreatment for 10 min. Each value represents the mean ± SEM (*n* = 3–4). ** *p* < 0.01 and *** *p* < 0.001 represent significant differences from the respective controls.

**Figure 3 pharmaceutics-14-02374-f003:**
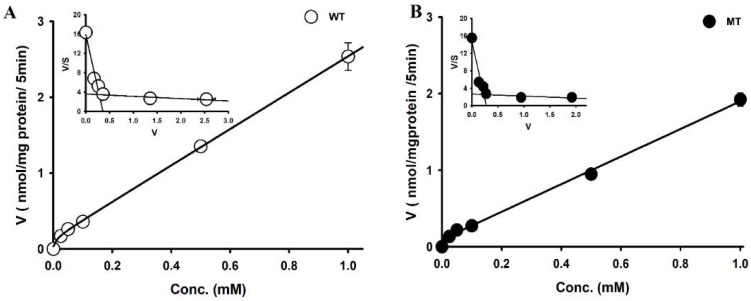
Saturation kinetics of [^3^H]choline uptake in NSC-34 cell lines. Uptake was carried out in either the presence or absence of cold choline at varying concentrations (0–1 mM) under physiological pH (7.4) for 5 min at 37 °C in (**A**) WT and (**B**) MT cell lines. Eadie–Hofstee plots are shown as insets. Each point represents the mean ± S.E.M (*n* = 3–4).

**Figure 4 pharmaceutics-14-02374-f004:**
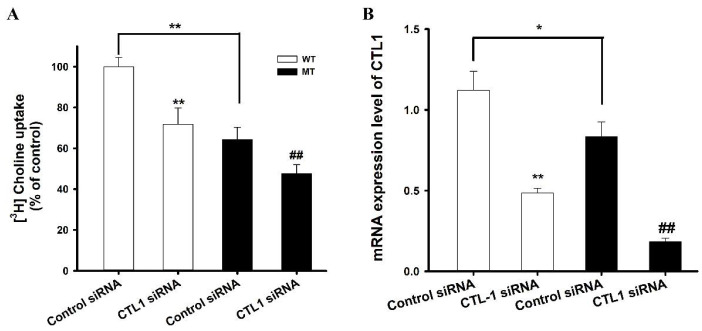
Relative expressions of transporters in ALS model cell lines. (**A**) [^3^H]Choline uptake after Slc44a1 (CTL1siRNA) transfection was performed at pH 7.4 and 37 °C for 5 min. (**B**) mRNA expression of CTL1 was analyzed after gene knockdown with Slc44a1 siRNA. Each value represents the mean ± SEM. (*n* = 3–4). * *p* < 0.05, ** *p* < 0.01 indicates a significant difference with respect to the WT control. ## *p* < 0.01 indicates a significant difference with respect to the MT control.

**Figure 5 pharmaceutics-14-02374-f005:**
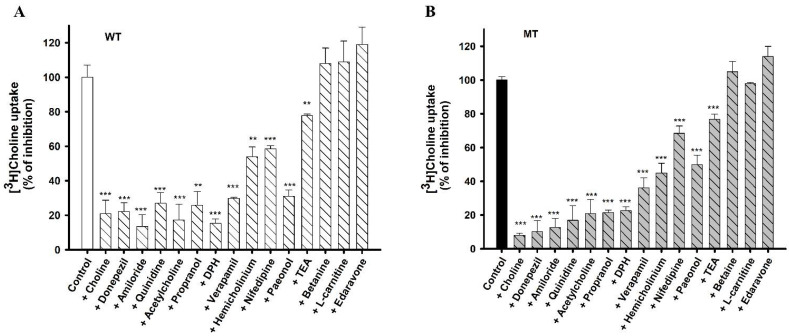
Effect of drugs on choline uptake in (**A**) WT and (**B**) MT cell lines. [^3^H]Choline uptake was analyzed for 5 min at 37 °C and pH 7.4 in the presence or absence of drugs (0–2 mM) in NSC-34 cell lines. Each data represents mean ± SEM (*n* = 3–4). ** *p* < 0.01, and *** *p* < 0.001 represent significant differences with respective controls. TEA, Tetraethyl ammonium.

**Figure 6 pharmaceutics-14-02374-f006:**
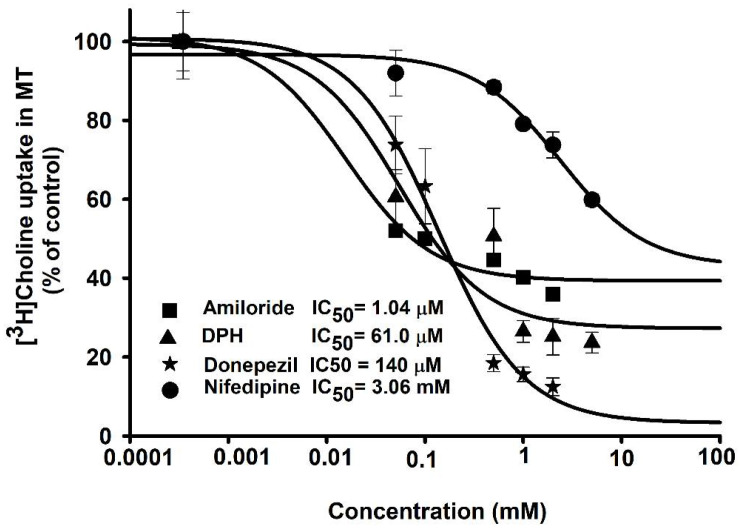
Inhibitory effects of drugs in a dose-response manner. Half inhibitory concentrations (IC_50_) were analyzed either in the presence or absence of amiloride, diphenhydramine (DPH), donepezil, and nifedipine at concentrations of 0–2 mM at pH 7.4 and 37 °C on the uptake of [^3^H]choline in the MT cell line. The data represents mean ± S.E.M (*n* = 3–4).

**Figure 7 pharmaceutics-14-02374-f007:**
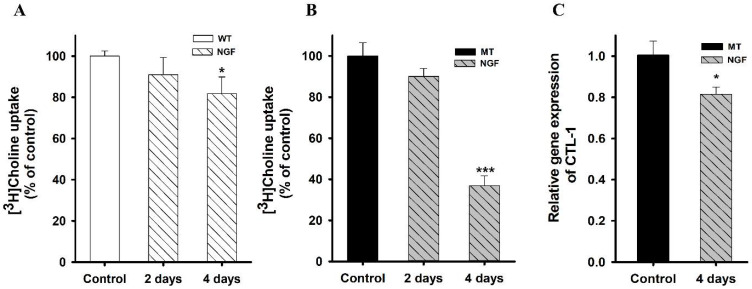
NGF pretreatment effect in ALS model cell lines. (**A**,**B**) [^3^H]Choline uptake was performed after treating WT and MT cell lines with 50 ng/mL NGF for 2 and 4 days. The uptake was carried out at pH 7.4 and 37 °C. (**C**) mRNA expression of CTL1 was analyzed after treating cells with NGF for 4 days in the MT cell line. The data represents mean ± S.E.M (*n* = 3–4). * *p* < 0.05 and *** *p* < 0.001, significantly different from the respective controls.

**Figure 8 pharmaceutics-14-02374-f008:**
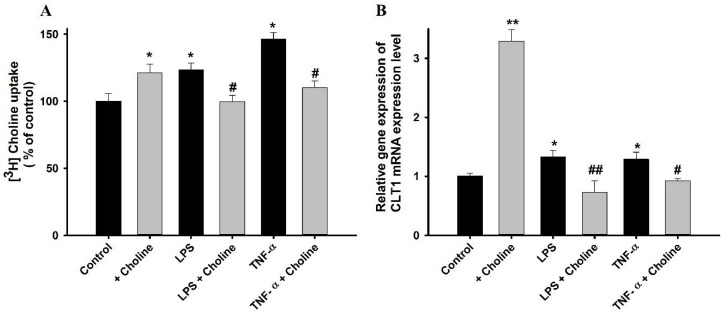
Pretreatment effects of pro-inflammatory cytokines in the ALS disease model cell line. (**A**) MT cell line was pretreated with 1 mM of choline, and 20 ng/mL of LPS and TNF-α either in the presence or absence of choline for 24 h. [^3^H]Choline uptake was carried out at a temperature of 37 °C and pH of 7.4 for 5 min. (**B**) Slc44a1 mRNA expression levels in MT cell line, after pretreatment with aforementioned pro-inflammatory cytokines. The data represents mean ± S.E.M (*n* = 3–4). * *p* < 0.05 and ** *p* < 0.01, significantly different from the control; # *p* < 0.05 and ## *p* < 0.01, significantly different from the respective controls, i.e., LPS and TNF-α.

**Figure 9 pharmaceutics-14-02374-f009:**
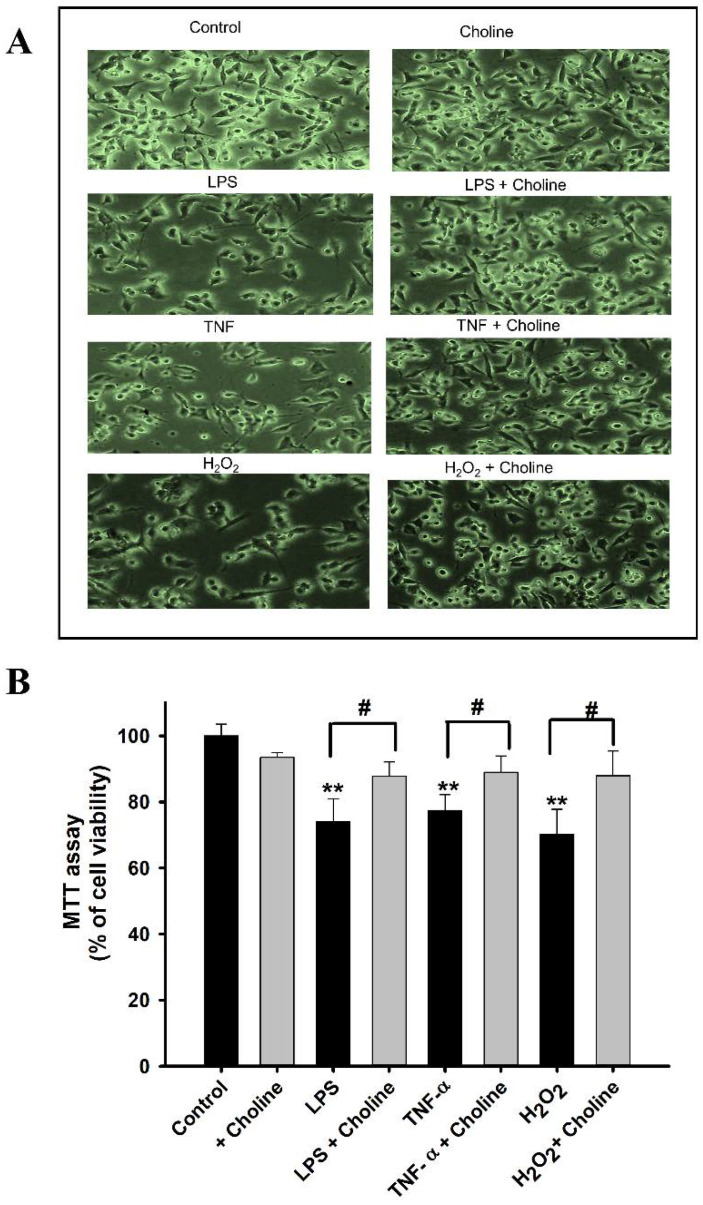
(**A**) Disease model cell line, images obtained via the EVOS XL cell imaging system (100×). (**B**) MTT (cell viability) assay was performed in the MT cell line upon treatment with pro-inflammatory cytokines and the oxidative agent H_2_O_2_ with or without the addition of choline (1 mM). Each data point represents mean ± S.E.M (*n* = 3–4). ** *p* < 0.01, significantly different from the control; # *p* < 0.05, significantly different from the respective controls, i.e., LPS, TNF-α, and H_2_O_2_.

**Table 1 pharmaceutics-14-02374-t001:** Kinetics of [^3^H]choline transport in ALS model cell lines.

Parameters	WT	MT
K_m1_ (µM)	10.1 ± 2.5	4.4 ± 0.64 *
K_m2_ (µM)	280 ± 21	43.7 ± 1.2 ***
V_max1_ (pmol/mg protein/min)	101 ± 20	55 ± 11 *
V_max2_ (pmol/mg protein/min)	107 ± 81	153 ± 16

K_m1_ and K_m2_ are the transport affinity values, and V_max1_ and V_max2_ are the maximum transport velocity values at high- and low-affinity sites, respectively. * *p* < 0.05, *** *p* < 0.001 indicates significant differences with respect to the WT control. Each value represents the mean S.E.M. (*n* = 3).

## Data Availability

All data included in this study.

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
