# Peer review of "Protective Effects of Choline against Inflammatory Cytokines and Characterization of Transport in Motor Neuron-like Cell Lines (NSC-34)"

_pharmaceutics, 2022, doi:10.3390/pharmaceutics14112374_

Round 1

Reviewer 1 Report

In article, Protective Effects of Choline against Inflammatory Cytokines 2 and Characterization of Transport in Motor Neuron-Like Cell 3 Lines (NSC-34), Latif et al, unravel the involvement of choline and choline transport system on the well being of neuronal cells in context of Amyotrophic Lateral Sclerosis. The authors utilize ALS cell models to demonstrate the positive effect of choline on the choline uptake itself in WT and MT cells. The authors also utilize the same assay as s surrogate to determine the effect multiple pharmacological drugs already in clinic with some of them already tested in case of ALS. The authors also show the direct involvement of CTL1 transport channel responsible of Choline uptake and finally the authors provide an account of effects of choline treatment against inflammatory cytokines and their effect on WT and MT cells survival. Overall, the authors make sincere efforts in highlighting the importance of choline transport in ALS using the in vitro model system. However, the manuscript does lack a clear link between the written script and the data presented and leave the reader confused with the take home message. Additional details and modification of the written portion will make this manuscript less ambiguous and more informative and thus bolstering the message of the study bringing the protective effects of choline to the light.

1. Based on the content in the introduction (line 44-line 52), the authors provide a convincing argument that choline acts as a neuroprotective agent. what is lacking in this paragraph is the main point being whether uptake of choline by the neuronal cells is a good thing or not. Meaning does ability to uptake choline indicator of good neuronal health. Similarly, in line 70-71, the authors mention that "in ALS patients, it has been demonstrated that the concentration of choline was lower compared with healthy controls". Does this mean that the overall low choline and lower choline uptake are deleterious for neuronal health in ALS? Without establishing this notion regarding the choline uptake, it is difficult to read through the data and comprehend its interpretation.

2. In methods 2.2, please provide the details on generation of WT and MT NSC-34 lines as well as any  information on the expression levels of WT and G93A hSOD1 in these cells.

3. IN line 145, the authors mention "uptake rate was reduced in the disease model cell line MT as compared with that of the 145 WT" which matches figure 1A. Doesn't this mean that lower choline uptake is deleterious for neuronal health? Thus, small molecules/drugs/treatments that improve the choline uptake will potential be beneficial in ALS?

4. Please replace the Table 1 with the actual graph showing the choline uptake between different conditions. It is difficult to understand the overall effect of any treatments as they are represented as relative uptake (% of control) considering there is already an inherent difference between the choline uptake between WT and MT. For example, with choline treatment supposedly increasing the choline uptake, does addition of +Choline 10mM results in only 7.85% uptake relative to control (which would be 92.15% reduction in uptake) or does it improve the uptake by 7.85% of control?

5. With the same ambiguity mentioned in point 4, the authors claim that removing Na+ decreased the choline uptake. does that mean addition of NMG and LiCl reduced the relative uptake to 39% of control? If that's what these number represent, please clarify why significantly higher effect is observed in WT cells than MT cells. Similarly, please clarify why disruption of membrane potential has significantly higher effect on MT cells as compared to WT cells (37% vs 61%).

Also, if these numbers actually reflect the reduced %, how does +choline treatment results in only 7% and 8% uptake in WT and MT, showing choline addition actually inhibiting the choline uptake which goes against the central hypothesis of this study.

6. Line 190, please include (Table 2) to point out the data being discussed in this point.

7. Please clarify why the Km2 difference of 280+/-21 in WT and 43.7 +/-1.2 in MT is not statistically significant?

8. Line 211. Please include (CTL1 siRNA) in parenthesis after Slc44a1 siRNA in the text as it is labelled in the figure.

9. Please provide if you have any rescue data on choline uptake after overexpression of CTL1 in NSC cells WT and MT.

10. It is not clear how authors chose which pharmacological compounds to test in the choline uptake assay. The authors do mention properties of these compounds but please provide additional information in the text regarding these compounds and what are they currently being used for in clinic if any.

11. Similar to Table 1, it is unclear if the data provided in this table talks about the % uptake compared to control or increase/decrease with respect to control. Please provide a graphical representation of the data.

12. The authors mention that some of the compounds tested have been shown to have neuroprotective effects in some cases even in ALS models. However, the data suggests that most of the compounds reduce the choline uptake in WT and MT cells. Considering choline uptake is a beneficial trait, is it to assume by the reader that most of these compounds would have deleterious effects on the neuronal health? Also, did the authors test any additional compounds that increase the choline uptake in WT and MT cells?

13. The authors state that NGF is neuroprotective and despite that pretreatment with NGF reduces the choline uptake as well as CTL1 expression. Authors thus suggest that the NSC-34 cells are sensitive to prolonged treatment to NGF. Please comment if this is a cell-specific phenomenon and if not, its implications in the clinic wit respect to NGF as a potential treatment candidate.

14. Considering the notion that higher choline uptake is beneficial (as seen by higher uptake in WT compared to MT in Fig.1), the data shows that pro-inflammatory cytokines LPS and d TNF-É‘ for 24 h resulted in a significant increase in the uptake of [3H]choline (Line 297-298 and fig 6). Please clarify why the proinflammatory cytokines have this effect on MT lines. 

15. Based on all the data till this point, the authors claim that addition of choline enhances the choline uptake however based on the data in fig 6, addition of choline after pretreatment with LPS and d TNF-É‘ actually reduces the choline uptake. Please clarify why addition of choline reduce the choline uptake.

16. The authors also claim that the expression of CTL1 was increased with the inflammatory states, and the addition of choline resulted in restoration, however, please clarify why the CTL1 expression is highest in choline treated cells compared to control and even LPS and d TNF-É‘ pretreated cells (Fig 6B) which does not match with the concurrent choline uptake rates in fig 6A.

Also, please clarify why the choline addition to LPS and d TNF-É‘ pretreated cells show even reduced CTL1 mRNA expression compared to just LPS and d TNF-É‘ pretreated cells. 

Considering increased choline uptake is beneficial which is in turn dependent on increased CTL1 expression (based on more in WT vs MT), this data set makes an ambiguous claim that cytotoxic effects of cytokines are due to increased uptake and increased CTL1.

17. Please provide a rescue experiment data as mentioned in point 9 in support of the claim that CTL1 transporter may help the sodium dependent transport of choline intro neurons.

18. Please provide the data supporting protective role of choline  against oxidative stress and excitotoxins.

Author Response

Reviewer 1

In article, Protective Effects of Choline against Inflammatory Cytokines 2 and Characterization of Transport in Motor Neuron-Like Cell 3 Lines (NSC-34), Latif et al, unravel the involvement of choline and choline transport system on the wellbeing of neuronal cells in context of Amyotrophic Lateral Sclerosis. The authors utilize ALS cell models to demonstrate the positive effect of choline on the choline uptake itself in WT and MT cells. The authors also utilize the same assay as s surrogate to determine the effect multiple pharmacological drugs already in clinic with some of them already tested in case of ALS. The authors also show the direct involvement of CTL1 transport channel responsible of Choline uptake and finally the authors provide an account of effects of choline treatment against inflammatory cytokines and their effect on WT and MT cells survival. Overall, the authors make sincere efforts in highlighting the importance of choline transport in ALS using the in vitro model system. However, the manuscript does lack a clear link between the written script and the data presented and leave the reader confused with the take home message. Additional details and modification of the written portion will make this manuscript less ambiguous and more informative and thus bolstering the message of the study bringing the protective effects of choline to the light.

  1. Based on the content in the introduction (line 44-line 52), the authors provide a convincing argument that choline acts as a neuroprotective agent. What is lacking in this paragraph is the main point being whether uptake of choline by the neuronal cells is a good thing or not. Meaning does ability to uptake choline indicator of good neuronal health. Similarly, in line 70-71, the authors mention, "in ALS patients, it has been demonstrated that the concentration of choline was lower compared with healthy controls". Does this mean that the overall low choline and lower choline uptake are deleterious for neuronal health in ALS? Without establishing this notion regarding the choline uptake, it is difficult to read the data and comprehend its interpretation.

Response: Thank you for your comment. We have added description in the manuscript

Line # 44-49 and Line # 74-79

  1. In methods 2.2, please provide the details on generation of WT and MT NSC-34 lines as well as any information on the expression levels of WT and G93A hSOD1 in these cells.

Response: The NSC-34 cell line is a fusion cell line derived from enriched spinal cord motor neurons and neuroblastoma. Mouse NSC-34 cells are stably expressing human superoxide dismutase 1 (hSOD1, WT), which is considered as control, and mutant NSC-34/hSOD1G93A (MT), cell line which overexpresses the human mutant SOD1 gene mutation with the substitution of glycine with alanine at position 93. This MT cell line displayed less differentiation and proliferation while mimicking the clinical circumstances linked to motor neuronal dysfunction in ALS. Therefore, NSC-34/hSOD1G93A MT is considered as disease model cell line

  1. IN line 145, the authors mention, "uptake rate was reduced in the disease model cell line MT as compared with that of the 145 WT" which matches figure 1A. Doesn't this mean that lower choline uptake is deleterious for neuronal health? Thus, small molecules/drugs/treatments that improve the choline uptake will potential be beneficial in ALS.

Response: We appreciate reviewers comment. In line 145, we are not mentioning the concentration of choline that is beneficial or deleterious to neuronal health. In Figure 1A, we are showing the uptake rate of choline transport in a time dependent manner. The reason is to find that carrier-mediated transporter is time dependent or not

In addition, the rate of choline transport in MT cell line (which mimics the motor neurons in ALS disease) is significantly low as compared to the WT therefore showing sensitivity to choline uptake as compared to the WT (control) cell line.

  1. Please, replace the Table 1 with the actual graph showing the choline uptake between different conditions. It is difficult to understand the overall effect of any treatments as they are represented as relative uptake (% of control) considering there is already an inherent difference between the choline uptake between WT and MT. For example, with choline treatment supposedly increasing the choline uptake, does addition of +Choline 10mM results in only 7.85% uptake relative to control (which would be 92.15% reduction in uptake) or does it improve the uptake by 7.85% of control?

Response: We have changed the table 1 to Figure 2 for more clarity.

  1. With the same ambiguity mentioned in point 4, the authors claim that removing Na+ decreased the choline uptake. does that mean addition of NMG and LiCl reduced the relative uptake to 39% of control? If that's what these number represent, please clarify why significantly higher effect is observed in WT cells than MT cells. Similarly, please clarify why disruption of membrane potential has significantly higher effect on MT cells as compared to WT cells (37% vs 61%).

Also, if these numbers actually reflect the reduced %, how does +choline treatment results in only 7% and 8% uptake in WT and MT, showing choline addition actually inhibiting the choline uptake which goes against the central hypothesis of this study.

Response: Here in Table 1, we determined the driving force and membrane potential for the transporter. To find that the carrier mediated transport system required ions for transporting choline in and out of the cell, the uptake was performed in the presence or absence of sodium.

Furthermore, the uptake of choline was performed in the presence of 10 mM of cold choline; it showed concentration dependency by more than 90% of strong inhibition (1000 times of the Km value; 10 mM) in both cell lines.

  1. Line 190; please include (Table 2) to point out the data being discussed in this point.

Response: Thank you for pointing this. We have moved the Table 2 position

  1. Please clarify why the Km2 difference of 280+/-21 in WT and 43.7 +/-1.2 in MT is not statistically significant?

Response: We highly appreciate reviewers’ comment. The significance mark was missing in the Table, we have corrected and added significance ***p< 0.001 in Table 1.

  1. Line 211. Please include (CTL1 siRNA) in parenthesis after Slc44a1 siRNA in the text as it is labelled in the figure.

Response: We have added the CTL1 siRNA in the parenthesis

  1. Please provide if you have any rescue data on choline uptake after overexpression of CTL1 in NSC cells WT and MT.

Response: We have shown in Figure 6, in the presence of inflammatory states, the overexpression of CTL-1 in MT disease model cell line.

  1. It is not clear how authors chose which pharmacological compounds to test in the choline uptake assay. The authors do mention properties of these compounds but please provide additional information in the text regarding these compounds and what are they currently being used for in clinic if any.

Response: In Table 3 (now as per reviewers comment changed to Figure 5). We have studied the analog inhibition study. Basic drugs that have structural similarity with choline studied to show they compete with the same binding site in the motor neurons as choline.

  1. Similar to Table 1, it is unclear if the data provided in this table talks about the % uptake compared to control or increase/decrease with respect to control. Please provide a graphical representation of the data.

Response: To avoid confusion, we have changed Table 2 to Figure 5.

  1. The authors mention that some of the compounds tested have been shown to have neuroprotective effects in some cases even in ALS models. However, the data suggests that most of the compounds reduce the choline uptake in WT and MT cells. Considering choline uptake is a beneficial trait, is it to assume by the reader that most of these compounds would have deleterious effects on the neuronal health? In addition, did the authors test any additional compounds that increase the choline uptake in WT and MT cells?

Response: The data shown with pharmacological compounds shows % of inhibition, as it does not mean that it have deleterious effect on neuronal health, but it indicate that these compounds shows either competitive /non-competitive inhibition with choline and they may follow the same binding site or transporter as used by choline in NSC-34 cell lines.

  1. The authors state that NGF is neuroprotective and despite that pretreatment with NGF reduces the choline uptake as well as CTL1 expression. Authors thus suggest that the NSC-34 cells are sensitive to prolonged treatment to NGF. Please comment if this is a cell-specific phenomenon and if not, its implications in the clinic with respect to NGF as a potential treatment candidate.

Response: Nerve growth factor member of neurotrophic family and has critical role in the growth, survival and differentiation of peripheral and central nervous system neurons. Choline is also beneficial on neuron in the synthesis of acetylcholine. Therefore, we decided to study the co-treatment effect of choline and NGF in the ALS model cell lines, and to find out NGF pretreatment will increase or decrease the choline uptake and CTL1 level.

  1. Considering the notion that higher choline uptake is beneficial (as seen by higher uptake in WT compared to MT in Fig.1), the data shows that pro-inflammatory cytokines LPS and d TNF-É‘ for 24 h resulted in a significant increase in the uptake of [3H]choline (Line 297-298 and fig 6). Please clarify why the proinflammatory cytokines have this effect on MT lines.

Response: Earlier study has shown that inflammatory mediators are elevated in the disease state and leads to inflammatory responses, which relates to pathology of disease.

In our study, the uptake of choline increases with the LPS and TNF-É‘. In order to find the reason for the increase in the uptake we performed mRNA expression of CTL-1. The results showed that in the presence of inflammatory states the expression of CTL1 mRNA was increased, thus confirming our uptake results.

  1. Based on all the data until this point, the authors claim that addition of choline enhances the choline uptake however based on the data in fig 6, addition of choline after pretreatment with LPS and d TNF-É‘ actually reduces the choline uptake. Please clarify why addition of choline reduce the choline uptake.

Response: We have shown in Figure 6 that in the presence of inflammatory states, the uptake of choline is reduced. The exact mechanism for this reduction is unknown. However, the data shown in Figure 6B, the reduction in the CTL-1 might be the reason for lower uptake of choline.

  1. The authors also claim that the expression of CTL1 was increased with the inflammatory states, and the addition of choline resulted in restoration, however, please clarify why the CTL1 expression is highest in choline treated cells compared to control and even LPS and d TNF-É‘ pretreated cells (Fig 6B) which does not match with the concurrent choline uptake rates in fig 6A.

Also, please clarify why the choline addition to LPS and d TNF-É‘ pretreated cells show even reduced CTL1 mRNA expression compared to just LPS and d TNF-É‘ pretreated cells. 

Considering increased choline uptake is beneficial which is in turn dependent on increased CTL1 expression (based on more in WT vs MT), this data set makes an ambiguous claim that cytotoxic effects of cytokines are due to increased uptake and increased CTL1.

Response: We appreciate reviewers comment. In Figure 6A and 6B, we have maintained the same inflammatory conditions, the addition of choline alone showed the protective effect in diseases model cell line MT, in which the expression of CTL-1 was lower as compared to WT (as shown in Figure 3). Therefore, we concluded that addition of choline have protective effect. In case of addition of choline to inflammatory states, choline resulted in reducing the levels to normal. Similar to our results, the uptake of choline was increased by LPS in microglia (1)

Reference

Okada, T.; Muto, E.; Yamanaka, T.; Uchino, H.; Inazu, M. Functional Expression of Choline Transporters in Microglia and Their Regulation of Microglial M1/M2 Polarization. Int. J. Mol. Sci. 2022, 23, 8924, doi:10.3390/ijms23168924.

  1. Please, provide a rescue experiment data as mentioned in point 9 in support of the claim that CTL1 transporter may help the sodium dependent transport of choline intro neurons.

Response: Choline is transported by different transporters in various cells. Main choline transporter CHT is sodium dependent, whereas choline like transporter (CTL1) is sodium independent. However, in our study, there was no expression of CHT in NSC-34 cell lines. Whereas, CTL-1 was expressed. In other cell lines, CTL-1 is sodium independent, however, in our NSC-34 cell lines, the transport of choline by CTL-1 is sodium dependent.

  1. Please, provide the data supporting protective role of choline against oxidative stress and excitotoxins.

Response: Animal data has shown that supplementation of choline administration to healthy rats resulted in beneficial effect on cognitive and locomotor performances (1). In addition, research on a mouse model for pain has reported that, by stimulating the nicotinic receptor a7, choline decreases TNF release from the macrophages, and choline appears to be effective in the inhibition of inflammatory cytokines (2). Previous research has reported the antioxidant effect of choline in immune organs by regulating the antioxidant system and decreasing oxidative damage (3).

  1. Tabassum, S.; Haider, S.; Ahmad, S.; Madiha, S.; Parveen, T. Chronic choline supplementation improves cognitive and motor performance via modulating oxidative and neurochemical status in rats. Pharmacol. Biochem. Behav. 2017, 159, 90–99, doi:10.1016/j.pbb.2017.05.011.
  2. Rowley, T.J.; McKinstry, A.; Greenidge, E.; Smith, W.; Flood, P. Antinociceptive and anti-inflammatory effects of choline in a mouse model of postoperative pain. Br. J. Anaesth. 2010, 105, 201–207, doi:10.1093/bja/aeq113.
  3. Wu, P.; Jiang, W.D.; Liu, Y.; Chen, G.F.; Jiang, J.; Li, S.H.; Feng, L.; Zhou, X.Q. Effect of choline on antioxidant defenses and gene expressions of Nrf2 signaling molecule in the spleen and head kidney of juvenile Jian carp (Cyprinus carpio var. Jian). Fish Shellfish Immunol. 2014, 38, 374–382, doi:10.1016/j.fsi.2014.03.032.

Reviewer 2 Report

The novelty of the manuscript is good. The Protective Effects of Choline against Inflammatory Cytokines have been studied by diverse authors.
The technical quality of the manuscript is good and it does not need improvement before publication. The manuscript should be extended in scientific discussion. The authors presented their results and compared to some works, but did not present explanations for the reasons to reach these results.

Author Response

Reviewer 2

The novelty of the manuscript is good. The protective effects of choline against inflammatory cytokines have been studied by diverse authors. The technical quality of the manuscript is good and it does not need improvement before publication. The manuscript should be extended in scientific discussion. The authors presented their results and compared to some works, but did not present explanations for the reasons to reach these results.

Response: We are grateful to the reviewer for such comments. We have revised our manuscript.

Reviewer 3 Report

This paper by Sana Latif et al demonstrated the characteristics of choline transport in Motor neuron like cell line (NSC-34 cell)-based in vitro ALS model. They also showed that Choline had protective effects against inflammatory cytokines induced injury. The general purpose of this study is clear. The study appears to be of interest, whereas the experiments have several problems. In my opinion, therefore, this manuscript is not recommended for publication in its present form, but can accept as the paper after throughout revisions.

1. Post hoc test should be written in the methods or figure legends.

2. Retinoic acid (RA) has been shown to inhibit proliferation, promote neuritogenesis, and convert into motor neuron-like cells from proliferating NSC-34 cells (Johann et al., 2011; Maier et al., 2013; Petrozziello et al., 2017; Nango and Kosuge 2022) and into motor neurons from iPS cells (Qu et al., 2014; Shimojo et al., 2015). Therefore, RA is generally used as a differentiation factor in NSC-34 cells.

Why did you use NGF in this study?

Please describe the reason and show the evidence about NGF as a differentiation factor in NSC-34 cells in the revised manuscript.

3. Fig7A

The image quality of that is a little bad. I feel that those images are distorted in Fig7A. Please modified this point in the revised manuscript.

Author Response

Reviewer 3

This paper by Sana Latif et al demonstrated the characteristics of choline transport in Motor neuron like cell line (NSC-34 cell)-based in vitro ALS model. They also showed that Choline had protective effects against inflammatory cytokines induced injury. The general purpose of this study is clear. The study appears to be of interest, whereas the experiments have several problems. In my opinion, therefore, this manuscript is not recommended for publication in its present form, but can accept as the paper after throughout revisions.

  1. Post hoc test should be written in the methods or figure legends.

Response: Description is added in the methods part under the section 2.6: Kinetic parameter estimation of [3H]choline uptake in NSC-34 cell lines.

  1. Retinoic acid (RA) has been shown to inhibit proliferation, promote neuritogenesis, and convert into motor neuron-like cells from proliferating NSC-34 cells (Johann et al., 2011; Maier et al., 2013; Petrozziello et al., 2017; Nango and Kosuge 2022) and into motor neurons from iPS cells (Qu et al., 2014; Shimojo et al., 2015). Therefore, RA is generally used as a differentiation factor in NSC-34 cells.

Why did you use NGF in this study?

Please describe the reason and show the evidence about NGF as a differentiation factor in NSC-34 cells in the revised manuscript.

Response: Thank you for the comment. In our manuscript, we have used NGF, as it previous study has shown that differentiation by various combinations of cAMP, retinoic acid and nerve growth factors (NGF) might provide cells of different morphologic maturity as well as activities of acetylcholine and acetyl-CoA metabolism (1). As, choline and NGF have beneficial effect for the neurons, so we wanted check the change of choline uptake with the pretreatment of NGF on the uptake of choline, and checked the reason of change with transporter mRNA level. Similar to our results, NGF has shown to increase significantly the level of choline incorporation into phosphatidylcholine in PC12 cells at 4 days at the concentration of 50 ng/ml (2).

Furthermore, NSC-34 cell lines are suitable for the study nerve growth factor as shown in earlier study addressing the use of motor neuron like NSC-34 cell lines for the investigation of neurotrophin receptor trafficking and associated subcellular processes (3).

  1. Szutowicz, A. Aluminum, NO, and nerve growth factor neurotoxicity in cholinergic Neurons. J. Neurosci. Res. 2001, 66, 1009–1018, doi:10.1002/jnr.10040.
  2. Araki, W.; Wurtman, R.J. Control of membrane phosphatidylcholine biosynthesis by diacylglycerol levels in neuronal cells undergoing neurite outgrowth. Proc. Natl. Acad. Sci. U. S. A. 1997, 94, 11946–11950, doi:10.1073/pnas.94.22.11946.
  3. Dusan Matusica, Matthew P. Fenech, Mary-Louise Rogers, and R.A.R. Characterization and Use of the NSC-34 Cell Line for Study of Neurotrophin Receptor Trafficking. J. Neurosci. Res. 2007, 3253, 3244–3253, doi:10.1002/jnr.
  4. Fig7A

The image quality of that is a little bad. I feel that those images are distorted in Fig7A. Please modified this point in the revised manuscript.

Response: Thank you for your suggestion. We have added modified figure 7A in the manuscript
